# Cannabigerol Alleviates Liver Damage in Metabolic Dysfunction-Associated Steatohepatitis Female Mice via Inhibition of Transforming Growth Factor Beta 1

**DOI:** 10.3390/nu17091524

**Published:** 2025-04-30

**Authors:** Raznin Joly, Fariha Tasnim, Kelsey Krutsinger, Zhuorui Li, Nicholas A. Pullen, Yuyan Han

**Affiliations:** 1College of Medicine, University of Cincinnati, Cincinnati, OH 45267, USA; jolyra@mail.uc.edu; 2Department of Biological Sciences, University of Northern Colorado, Greeley, CO 80639, USA; fariha.tasnim@unco.edu (F.T.); kelsey.krutsinger@unco.edu (K.K.); nicholas.pullen@unco.edu (N.A.P.); 3College of Biology, China Agricultural University, Beijing 100107, China; lizhuorui0@gmail.com

**Keywords:** metabolic dysfunction-associated steatotic liver disease (MASLD), metabolic dysfunction-associated steatohepatitis (MASH), cannabigerol, transforming growth factor beta 1 (TGF-β1)

## Abstract

**Background and Aims:** Metabolic dysfunction-associated steatohepatitis (MASH), a progressive form of metabolic dysfunction-associated steatotic liver disease (MASLD), involves inflammation, fibrosis, steatosis, and oxidative stress. Previous research from our lab shows that cannabigerol (CBG) reduces inflammation and fibrosis in male MASH mice, but its effects in females remain unknown. Given immune cell population changes in MASLD patients, this study examines CBG’s impact on methionine-choline deficient (MCD) diet-induced MASH in female mice. **Methods:** MCD-fed female mice are supplemented with two different doses for three weeks. Liver fibrosis, steatosis, oxidative stress, ductular reaction, and inflammation are assessed via Sirius Red, Oil Red O, immunohistochemistry, and immunofluorescence staining. Immune cell changes in non-parenchymal cells (NPCs) are analyzed via flow cytometry. **Results:** CBG treatment improves liver health by reducing leukocyte infiltration. Both CBG doses significantly decrease fibrosis, oxidative stress, ductular proliferation, and inflammation in MCD-fed mice, including monocyte and T lymphocyte reductions. Additionally, CBG downregulates mast cell activation, inhibiting transforming growth factor (TGF)-β1 release, thereby suppressing hepatic stellate cell activation. This reduces collagen deposition, fibrosis, and ductular proliferation. **Conclusions:** Our findings provide insights for pre-clinical and clinical research, highlighting CBG’s potential therapeutic role and dosage considerations in mitigating liver fibrosis and inflammation in female patients.

## 1. Introduction

The clinical management of metabolic dysfunction-associated steatohepatitis (MASH) is a growing concern due to its associated complications and overall mortality rate [1]. MASH is a progressive stage of metabolic dysfunction-associated steatotic liver disease (MASLD) and is mainly induced by an unhealthy diet or lifestyle. The primary histological indicators of MASH development from MASLD include hepatocellular ballooning, pericellular fibrosis, hepatic steatosis, oxidative stress, and lobular inflammation [2]. The prevalence of MASH in the United States has seen a notable increase over the last decade. One study found that the prevalence rate of MASH has increased from 1.5% to 2.79% in the last decade, with slightly higher incidence in females [3]. Other reports show that between 1.5% to 6.45% of patients with MASLD eventually develop MASH [1]. There are several risk factors for the progression of MASH from MASLD, including type 2 diabetes and obesity. The current Clinical Management I for MASH is limited and mainly focuses on weight loss, lifestyle changes, and medications that target the underlying causes of the disease [1]. Currently, Resmetirom is the only Food and Drug Administration (FDA)-approved medication available for MASH treatment, and it is very expensive with many side effects [4]. Uncontrolled MASH with advanced fibrosis leads to liver cirrhosis and hepatocellular carcinoma and eventually requires liver transplantation [5]. Therefore, there is an urgent need to explore drugs or supplements that can alleviate liver damage in MASH patients. In our study, the experimental models are female mice. Mice share about 80% of their genes with humans, making them a good model for studying MASH [6]. The observations from this study can serve as a guide for advancing human trials to confirm dosage, efficacy, and side effects. While mice are not perfect substitutes for humans, they provide valuable early-stage data that inform future clinical research and drug development.

Immune cell modulation is important in the pathogenesis of MASLD and MASH. Studies show changes in immune cell populations in MASLD patients, including increases in CD4^+^ T_h_2 cells and decreases in natural killer cells and CD3^+^ and CD8^+^ T cells [7]. Additionally, a recent study involving the MASH mouse model reveals five distinct macrophage populations [8]. These recruited macrophages (Ly6C^+^) are enriched around the hepatic portal and central vein, where they are commonly associated with fibrosis in the cirrhotic liver. In addition, researchers reveal increased mast cell activation with histamine release, thus promoting biliary senescence, hepatocyte steatosis, and the further activation of HSCs [9]. Along with other cytokines, mast cells secrete transforming growth factors (TGF-β1), which play a crucial role in fibrogenesis by activating HSC [10]. Overall, studies indicate that mast cells and other immune cells such as monocytes, macrophages, and T cells are potential targets for MASH treatment.

CBG is a precursor molecule of Cannabidiol (CBD). Both are non-psychoactive bioactive compounds in the cannabis plant (*Cannabis sativa* L.). Many preclinical studies with CBD reveal its therapeutic potential to reduce inflammation and oxidative stress in multiple pathological states, including colitis, diabetic complications, hepatic ischemia, and cardiomyopathies [11]. Recent phase 3 and 1 clinical trials have FDA approval for the evaluation of CBD’s effect in refractory childhood and glioblastoma multiforme, respectively [12,13]. While there are numerous studies related to the pharmacological action of CBD, not as much is known about CBG. While limited, in vitro and in vivo studies have shown that CBG has anti-inflammatory and antioxidant effects in neurological disease models [14,15]. Our previous research shows that a low dose of CBG (2.46 mg/kg/day) reduces inflammation and fibrosis in male MASH mice [16]. However, the study did not include female mice, making it challenging to fully understand the effect of CBG in MASH, especially for female patients. Additionally, to date, no research has examined the efficacy of CBG on liver oxidative damage, the modulation of hepatic immune cell populations, or the mechanism by which CBG targets different cell populations to inhibit MASH progression.

Therefore, this study aims to evaluate the role of CBG treatment in alleviating the symptoms related to MCD diet-induced MASH in female mice. We evaluate the effects of CBG in reducing fibrosis, oxidative stress, inflammation, ductular reaction, and steatosis in female MASH mouse models induced by the MCD diet. We also study the effect of CBG treatment on mast cell and HSC activation to understand the possible mechanism of action of CBG in reducing fibrosis, ductular reaction, and inflammation.

## 2. Materials and Methods

### 2.1. Diets, Reagents, and Antibodies

Control diet (CTR) (Catalog No. 94149) and MCD (Catalog No. 90262) are purchased from Envigo (Denver, CO, USA) [16], while all other reagents and antibodies are from Thermo Fisher Scientific (Denver, CO, USA), VWR (Radnor, PA, USA), BioLegend (San Diego, CA, USA), and Invitrogen (Fredrick, MD, USA) unless we otherwise indicate. CBG is purchased from Extract Lab (Broomfield, CO, USA). Antibodies and their source information are listed in Appendix A.

### 2.2. Animal Treatment

All proposed procedures are approved by the Institutional Animal Care and Use Committee at the University of Northern Colorado (protocol no. 2211CE-YH-M-2F, approval date: 8 July 2024). Seven-to-eight-week-old C57BL/6 female mice are housed in a 12 h light/dark cycle and given ad libitum access to the control diet (ctrl) or MCD diet. After two weeks of feeding these two different diets (CTR and MCD), mice are randomly divided into three subgroups within each diet, in which mice receive a low (2.46 mg/kg/day, denoted as LOW CBG) or a high (24.6 mg/kg/day, denoted as HIGH CBG) CBG dosage or vehicle (10 µL of tween-80 and 37.5 µL of dimethyl sulfoxide in 1.5 mL PBS) intraperitoneally three times per week for three weeks. Each group contains three to four mice. The high dose is one-tenth of the maximum amount of CBD that a human is allowed to consume in a day. This high human dose is then converted to the equivalent dose for mice [17]. The low dose was also selected based on information from previously published studies [18]. During the length of the experiment, body weights are measured weekly. At the end of week five, the mice are euthanized using sodium pentobarbital (50 mg/kg) with heparin (100 U/kg).

To test the impact of CBG on the immune cell populations of non-parenchymal cells (NPCs) in female MASH mice compared to the control, another set of female mice is randomly assigned to four groups (four mice per group): CTR, MCD, MCD + LOW CBG, and MCD + HIGH CBG. Five weeks after the treatment, liver perfusion is performed according to an established method [19] and NPCs are isolated to study the immune cell population via flow cytometry.

### 2.3. Histological Staining

In this study, 4.5 µm formalin-fixed paraffin-embedded (FFPE) liver slides are placed in a slide warmer at 65 °C until the paraffin is melted. The slides are then placed in two changes in xylene, followed by 100% ethanol, then 95%, and lastly 75% for 2 min each (e.g., deparaffinization). The slides are rinsed with 2–3 changes in tap water for 30–50 dips (e.g., rehydration). After the staining process, slides are dehydrated in 70%, 90%, and 100% ethanol. Finally, tissues are placed in xylene and mounted in mounting media. Afterward, images of the tissues are taken using a light microscope. These images are analyzed using ImageJ software (Version 1.53e).

### 2.4. Hematoxylin and Eosin Staining

To evaluate overall liver health, hematoxylin and eosin (H&E) staining is performed. After deparaffinization and rehydration, the tissues are stained with hematoxylin for 2 min and washed in ammonia water. They are then mordant in 95% ethanol followed by staining with eosin for 1 min. Following dehydration and mounting, all images for analysis are taken at 100× magnification.

### 2.5. Sirius Red Staining

Collagen deposition in liver tissues is measured with Sirius Red staining in order to evaluate liver fibrosis. The paraffin-embedded tissues are sliced, deparaffinized, and rehydrated as described in the Section 2.3. Then, the cell nuclei are stained with hematoxylin for 40 s, followed by rinsing in tap water. Next, the tissues are stained for collagen deposition with Picro-Sirius Red for one hour, followed by two washes in 0.5% acidified water. After that, they are dehydrated and mounted for analysis. All images are taken at 100× magnification.

### 2.6. Oil Red O Staining

To evaluate the lipid accumulation, 14 µm frozen liver sections are fixed in 10% neutral buffered formalin (NBF) for 10 min and then washed in 3 changes in water. Tissues are then placed in 100% propylene glycol for 5 min, followed by preheated Oil Red O staining for 8 min. After that, they are placed in 85% propylene glycol for 5 min and washed in 2 changes in distilled water. Further, tissues are counterstained with hematoxylin for 40 s and washed in running tap water for 3 min. Last, tissues are mounted in aqueous mounting media. All images are taken at 100× magnification.

### 2.7. Immunofluorescence Staining

Moreover, 14 µm frozen liver sections are fixed in 10% NBF for 10 min and then washed with cold PBS. The sections are then blocked for 20 min with 10% normal goat serum diluted in PBS. Thereafter, the sections are incubated in primary antibody overnight at 4 °C or approximately 1–2.5 h at room temperature. The next day, the slides are washed with PBS and stained with the secondary antibody for 45 min at room temperature. Finally, tissues are mounted with DAPI (4′,6-diamidino-2-phenylindole) mounting medium and stored in the dark at −20 °C until analyzed with a Zeiss 700 confocal microscope, with a maximum of 5 days between staining and imaging. All images are taken at 200× magnification.

### 2.8. Immunohistochemistry Staining

After deparaffinization and rehydration, tissues are incubated with 3% hydrogen peroxide (H_2_O_2_) for 10 min to block endogenous peroxide binding. Slides are then rinsed with tap water for 5 min and heated in citrate buffer in a high-pressure cooker (around 12 psi) for 30 min for antigen retrieval. The optimal antigen retrieval time is identified after testing at different times for each antibody. We recommend following common titration methods to identify the optimal antibody concentration for each target in addition to testing different antigen retrieval times. Tissues are incubated with blocking serum for one hour. Primary antibody diluted in 10% goat serum in PBS is then added. After overnight incubation at 4 °C, a secondary antibody diluted similarly is added. The tissues are then incubated for 30 min. After a wash with PBS, the tissues are incubated for 30 min with a priorly 20 °C-acclimated VECTASTAIN ABC reagent (Vector Laboratories, Newark, CA, USA). DAB reagent is added following PBS wash, and slides are monitored until color develops. Before mounting, tissues are counterstained with hematoxylin to highlight nuclei.

### 2.9. Isolation and Analysis of Non-Parenchymal Cells

Livers are perfused according to previously published methods immediately after euthanasia [19]. NPCs are then pelleted at 400× *g* for 5 min at 4 °C. Any remaining erythrocytes are cleared using ACK lysis buffer. Subsequently, the cell suspensions are aliquoted to achieve the desired cell concentration, which is typically around 10^6^ cells/mL per test, for staining. Specifically, 200 µL cell suspensions are stained for cell markers for 20 min on ice after blocking with a blocking antibody (CD16/32) for 10 min. After the 10 min incubation, 800 µL of incubation buffer is added to the cells, followed by centrifugation at 400× *g* for 5 min. The cells are then washed with incubation buffer three times before being analyzed on an Attune NxT cytometer (ThermoFisher, Waltham, MA, USA). The gates for different cell populations are obtained by using isolated NPCs from mouse liver fed with a regular diet. Antibodies against CD3, CD4, CD8a, CD14, CD45, CD16, CD115, CD11b, and Ly6C are utilized to differentiate different immune cell populations (see Appendix A for the antibody list). Flow data are analyzed using FCS Express 6 software.

### 2.10. Statistical Analysis

All data are analyzed using GraphPad Prism 9 software and reported as mean ± SEM (standard error mean). A one-way analysis of variance (ANOVA) is performed to test for significant differences between group means for each experiment in this study. Tukey’s post hoc tests were performed, comparing every pair of groups to identify significant relationships when significance is detected by the ANOVA. For all tests used, a *p*-value ≤ 0.05 is considered significant.

## 3. Results

### 3.1. Cannabigerol Treatment Decreases Oxidative Stress and Improves Overall Liver Health in Methionine–Choline-Deficient Diet-Fed Female Mice

Hepatic oxidative stress and impaired mitochondrial function significantly contribute to the onset and advancement of MASH. Oxidative stress, characterized by an imbalance between the generation of reactive oxygen species (ROS) and the capacity of the endogenous antioxidant system to counteract them, is a central factor of MASH pathogenesis [20]. Elevated ROS levels lead to oxidative changes in cellular components like DNA, lipids, and proteins, causing an accumulation of damaged molecules and contributing to liver damage. Evaluating the levels of those damaged biomolecules is a fundamental approach to gauging the level of oxidative stress [21]. We use immunofluorescence staining in frozen liver sections to assess the expression levels of 8-hydroxy-2′-deoxyguanosine (8-OHdG), a marker for DNA damage products. Our findings show minimal to no expression of 8-OHdG (shown as dark pink marked by white arrows in Figure 1A,B) in control diet-fed female mice, with or without CBG treatment. Conversely, female mice fed with the MCD diet exhibit pronounced oxidative stress (Figure 1A,B), which shows significantly diminished levels of 8-OHdG following treatment with both low and high doses of CBG injections.

An increased accumulation of leukocytes (Figure 1C) is observed near the hepatic arteries, portal vein, and bile ducts in the MCD-diet-induced MASH mice livers, whereas no clusters of leukocytes are observed in the control diet-fed mice groups. Meanwhile, both low and high doses of CBG treatment show improved liver health, evidenced by a decrease in leukocyte infiltration (Figure 1C). We also highlight a high amount of hepatocellular ballooning (Figure 1C) in the liver tissue sections from the MCD diet-fed mice groups, which is absent in control groups. Hepatocyte ballooning is a distinctive sign indicating cellular damage with ongoing inflammation and is characterized by enlarged and swollen liver cells, with or without Mallory–Denk bodies present in the cell’s cytoplasm [22]. We show little or almost no improvement in the hepatic ballooning when treated with low and high CBG (Figure 1C). In addition, CBG treatment does not recover the drop in liver-to-body weight ratio, a side effect that is not typically observed in human MASH patients.

### 3.2. Treatment with Cannabigerol Attenuates Hepatic Fibrosis and Ductular Proliferation in Metabolic Dysfunction-Associated Steatohepatitis Female Mice

The balance between extracellular matrix (ECM) production and degradation is disrupted during chronic liver damage, leading to the accumulation of ECM surrounding the hepatic triad region. Since the increased deposition of fibrous ECM proteins (e.g., collagen, elastin, laminin, and fibronectin) is a hallmark of fibrosis, liver tissues are stained for collagen with micro-Sirius red staining to evaluate fibrosis in MCD diet-fed female mice with and without CBG treatment. We show enhanced fibrosis (Figure 2A) in the MCD-fed mice group compared to the control group; both low and high doses of CBG administration reduce fibrotic tissue in MCD-fed mice liver (Figure 2A,B).

HSCs, major producers of ECM proteins, are recognized as key contributors to fibrogenesis. During chronic inflammation, these cells transition from a quiescent state to an activated state and adopt a proliferative myofibroblast phenotype. This shift is responsible for the primary production and deposition of collagen, a major component of fibrous tissue structure [23]. A strong connection between the liver fibrosis stage and enlarged HSCs is observed in obese patients, suggesting enlarged HSCs may be a new marker for metabolic liver disease progression [24]. To investigate the potential antifibrotic effects of CBG, we explore how CBG treatment influences hypertrophied HSCs. Using immunofluorescence staining, we examine tissue samples for the presence of desmin, an intermediate filament found in all HSCs and upregulated after activation. Hypertrophied HSCs from other quiescent HSCs are differentiated based on the morphological differences in combination with desmin [24]. We show a significant reduction in hypertrophied HSCs (Figure 2C,D) in the liver of both low- and high-CBG-treated female MASH mice compared to vehicle-treated female MASH mice.

An investigation involving individuals with MASH/ MASLD reveals substantial biliary ductular reaction in patients with MASH who have both portal and linking fibrosis—in contrast to individuals with only liver steatosis. Furthermore, the degree of this ductular reaction intensifies as the fibrosis stage advances [25]. Therefore, to further investigate whether CBG inhibits fibrosis via the inhibition of cholangiocyte proliferation (ductular reaction), we employ immunohistochemistry staining on FFPE liver tissues to target cytokeratin 19 (CK19), a biomarker protein that is specifically expressed in cholangiocytes in the liver. The results highlight a notable increase in ductular reaction (shown as dark brown in Figure 2E) in mice fed with the MCD diet compared to the control group. Upon administering CBG, we show a significant reduction in cholangiocyte proliferation compared to that of the MCD diet group (Figure 2E,F).

### 3.3. Cannabigerol Treatment Reduces Liver Inflammation in Metabolic Dysfunction-Associated Steatohepatitis via Inhibition of Infiltration of Monocytes and T-Lymphocytes

Liver inflammation, promoting fibrosis and ultimately resulting in cirrhosis, is an important sign that distinguishes MASH from MASLD [26]. Therefore, the inhibition of inflammation is a crucial strategy for MASH treatment. In addition, as MASH progresses, oxidative stress within the liver leads to the activation of pro-inflammatory cytokines [27]. Therefore, it is important to examine whether the bioactive molecule CBG shows an anti-inflammatory effect in MCD-induced MASH mouse model. The initial assessment of liver tissue inflammation involves evaluating the expression of CD45, a biomarker for white blood cells (leukocytes). In Figure 3A,B, there is a significant increase in CD45^+^ cells in the MCD groups compared to the control groups. In contrast, both CBG doses exhibit a significant reduction in CD45 infiltration in the MCD-fed mice. These results are consistent with those observed in the H&E stains.

Macrophages constitute the predominant population of immune cells in the liver and hold vital responsibilities in instigating liver inflammation. The liver contains two types of macrophages: liver-resident macrophages, known as Kupffer cells and macrophages derived from monocytes (generally referred to as just macrophages). Among these, monocyte-derived macrophages are one of the major promotors of liver inflammation. These monocyte-derived macrophages trigger inflammation and fibrogenesis via autocrine and paracrine activation and further attract leukocytes and HSCs, fostering complex intercellular communication. They also cause the transformation of HSCs into collagen-producing myofibroblasts through the secretion of TGF-β1 and platelet-derived growth factor [26]. It is shown that the infiltration of Ly6C^+^ monocytes is a crucial pathological occurrence that drives the development of steatohepatitis and, consequently, the progression of fibrosis in MASH [8]. Considering this, we examine whether CBG treatments reduce the recruitment and activation of monocyte-derived macrophages during MASH-induced liver damage. As shown in flow cytometric analysis (Figure 3C–F), there is a significant increase in the proportion of monocytes in the liver of MCD diet-fed mice. Specifically, the relative number of CD11b^+^, Ly6C^+^, and CD115^+^ cells are, on average, 5.4-fold, 13-fold, and 14-fold higher, respectively, in MASH-induced mice compared to the controls. Treatment with a low dose of CBG results in reductions of approximately 1.47-fold, 1.67-fold, and 2.2-fold in CD11b^+^, Ly6C^+^, and CD115^+^ cells, respectively, compared to the MCD diet-fed group. High-dose CBG treatment shows a modest 1.1-fold reduction in CD11b^+^ cells but shows more significant reductions in Ly6C^+^ and CD115^+^ cells, averaging approximately 1.3-fold and 1.5-fold, respectively, compared to the MCD diet-treated mice.

Since CBG inhibits leukocyte infiltration in the MASH liver, we would like to know whether it is caused by T cell population change. Following the flow cytometric analysis of isolated NPCs, we observe a noteworthy increase in the CD3^+^ cell population in MCD-fed mice, specifically CD4^+^, CD8a^+^, and CD4^+^/CD8a^+^ cell subsets, with rises of 5.7-fold, 6.79-fold, and 9.36-fold, respectively, compared to the control. Treatment with both low and high doses of CBG results in a reduction in total CD3^+^ cells by 1.84-fold and nearly 1.36-fold, respectively. Additionally, treatment with the low CBG dosage causes reductions of 1.96-fold, 1.77-fold, and 2.02-fold in the CD4^+^, CD8a^+^, and CD4^+^/CD8a^+^ cell populations within the liver. Meanwhile, the high CBG dosage leads to reductions of 1.35-fold, 1.53-fold, and 1.5-fold in the same cell subsets (Figure 3G–K).

### 3.4. Cannabigerol Downregulates Mast Cell-Secreted Transforming Growth Factor-β1 to Inhibit Hepatic Stellate Cell Activation, Resulting in Attenuation of Fibrosis

These outcomes concerning the functional efficacy of CBG in countering fibrosis and ductular proliferation in MASH motivate us to explore the mechanism through which CBG mitigates these symptoms. Building upon the findings of heightened mast cell activity in MASLD and MASH [28], we investigate the influence of mast cells in our female mouse MCD diet-induced model of MASH and assess the effectiveness of CBG in reducing hepatic mast cell activation and the subsequent release of pro-inflammatory and pro-fibrotic cytokines. We analyze mast cells in our experimental groups by staining for tryptase. Beta-tryptase, a neutral serine protease in mast cell granules, plays an important role in mediating the inflammatory process. The release of beta-tryptase can be used as a biomarker for mast cell degranulation and serve as a signal for mast cell activation [10]. Within the liver, mast cells are found to be increased in connective tissue around the hepatic triad. Both low and high doses of CBG treatment decrease the activation and accumulation of mast cells in the liver tissue of mice fed an MCD diet as shown by the tryptase expression (Figure 4A,B). No accumulation of mast cells is found near the hepatic triad in the control mice (Figure 4A).

Upon noticing the decrease in HSCs in CBG-treated mice (Figure 2C), we explore if both mast cells and HSCs undergo activation within the same periportal areas where a fibrotic scar is typically observed (Figure 2A). This is performed through immunohistochemistry double staining for mast cells and HSCs on liver tissue samples using tryptase and desmin as biomarkers, respectively. While the outcome displays the limited co-localization of mast cells and HSCs, both were situated near the hepatic portal triad and central vein. Further, we observe a reduction in both mast cell and HSC levels following both low and high doses of CBG treatment (Figure 4C).

TGF-β1 serves as a pro-inflammatory cytokine that initiates and promotes fibrogenesis within the liver [29]. During liver injury, infiltrated mast cells accumulate around the hepatic portal area and secrete TGF-β1, which promotes hepatic fibrosis by activating HSCs. Here, we evaluate the co-localized expression of FcεRI and TGF-β1 in the areas near the hepatic arteries, portal veins, and bile ducts by immunofluorescence to verify whether TGF-β1 is secreted from mast cells during fibrogenesis and inflammation. FcεRI functions as a cell surface receptor specifically for immunoglobulin IgE, serving as the initial trigger for the regulation and initiation of mast cell activation. As shown in Figure 4E, there is an increase in the co-localization of FcεRI and TGF-β1 (Figure 4E) in the MCD-fed mice, which is significantly reduced upon treatment with either a low or high dosage of CBG. The minimal or no co-expression of both markers is found in the control groups (Figure 4E,F). We also examine the colocalization of TGF-β1 and desmin (a marker for HSCs) to understand the effect of CBG on the TGF-β1-driven activation of HSCs. Likewise, an increase in the colocalized TGF-β1 and desmin is observed in the MCD diet-fed group, and a decrease is noted with both low and high dosages of CBG treatment in MASH mice (Figure 4D).

### 3.5. Cannabigerol Administration Causes Little or No Change in Lipid Accumulation

The MCD diet induces changes in hepatic lipid metabolism, potentially resulting in an excessive buildup of lipids and the development of a steatotic liver. The effect of CBG interventions is evaluated for lipid accumulation (shown as red in Appendix A) in female MASH mice by Oil Red O staining. The results show little or no change in the level of lipid accumulation in the liver when administered with CBG (Appendix A), similar to the observation in male MASH mice [16].

## 4. Discussion

MASH without management can progress into liver cirrhosis and liver cancer. MASH is now the second most common reason for liver transplant in females [5]. Our findings show that CBG effectively ameliorates fibrosis, inflammation, and oxidative stress in female MASH mice. Additionally, hepatic ductular reactions and mast cell activation associated with MASH are significantly reduced following CBG treatment via the inhibition of TGF-β1-induced HSCs activation. Furthermore, CBG administration inhibits monocyte-derived macrophages (CD11B^+^ and Ly6C^+^) and T lymphocytes in the liver of female MASH mice.

In MASH, an increase in hepatic oxidative stress and DNA damage from oxidation occurs alongside reduced antioxidant defenses and heightened inflammation [20]. Previous research has emphasized the antioxidative effect of drugs like Nobiletin while elucidating the potential therapeutic intervention to heal MASH-associated liver damage [30]. Our findings support the antioxidative effects seen in other studies regarding the therapeutic potential of CBG [14,15]. The cause of oxidative damage in MASH is largely attributed to mitochondrial dysfunction and the disruption of endoplasmic reticulum redox balance resulting from hepatic lipotoxicity [20]. However, the exact molecular mechanisms responsible for oxidative stress and the resulting liver damage are still not fully understood. A further exploration into the pharmacological pathways through which CBG reduces oxidative stress in MASH is of the utmost importance.

Research indicates that hepatic fibrosis plays a crucial role in determining the mortality of MASH patients, prompting a focus on identifying efficient and safe therapeutic agents capable of reducing fibrotic scarring in the liver [31]. We find that both low and high doses of CBG treatment could inhibit fibrosis via the inhibition of HSC activation. Although the optimal dose for female patients is still unclear as both dosages showed therapeutic effects, our data provide evidence that CBG could be a promising option for treating fibrosis in MASH. Activated HSCs play a critical role in ECM remodeling by secreting matrix metalloproteinase (MMP) enzymes, contributing to fibrogenic conditions [24]. One study shows that a significant decrease in MMP-9 expression and secretion by HSCs in fibrotic tissue is associated with hepatocellular carcinoma [32]. Thus, further assessing the impact of CBG on secreted MMP levels in MASH is crucial for understanding the molecular mechanisms involved in its antifibrotic pathophysiological role.

The progression from MASLD to MASH entails a variety of inflammatory processes mediated by residential and infiltrating immune cells. These cells turn the microenvironment into an inflammatory state, through the release of pro-inflammatory cytokines (such as tumor necrosis factors and interleukins) or in a direct contact manner [33]. It is shown that the interaction between immune cells and HSCs can stimulate fibrogenesis [26]. Limited evidence suggests a close communication between macrophages and HSCs or liver sinusoidal endothelial cells during MASH [33]. The increased infiltration of Ly6C^+^ monocytes into the liver is recognized as a critical factor in the progression of steatosis and fibrosis [34]. Several compounds targeting the chemokine receptors, such as CCR2/CCL2 and/or CCR5/CCL5, are under investigation in late-stage MASH patients. These treatments are being assessed with the aim of inhibiting monocyte infiltration in the liver. Furthermore, the recruitment of CD4^+^ and CD8^+^ T lymphocytes is also reinforced during the progression of MASH [34]. In this study, we show that CBG significantly reduces leukocyte infiltration, hepatic monocyte, and T lymphocyte populations in the MASH female mice. This further supports CBG as a potential novel treatment for alleviating MASH inflammatory responses. In general, MASH is characterized by an enhanced secretion of T_h_1-derived cytokines, such as interferon γ (IFNγ), CD8^+^ T cell-derived IFNγ and tumor necrosis factor α (TNFα), along with a reduced secretion of IL-4, IL-5, and IL-13 from Th2. Therefore, it is crucial to analyze the effect of CBG treatment on these pro-inflammatory and pro-fibrotic cytokines in female MASH mice.

The increased centrilobular ductular reaction is correlated with the escalation of necro/inflammatory activity and fibrosis stage [25]. Our findings regarding the inhibitory effects of CBG on bile duct hyperplasia and mast cell activation present novel avenues for research and treatment approaches to halt inflammation and fibrosis in MASH liver. Mast cells are present throughout the body but are often increased in liver damage, including MASH [10]. One study demonstrates the increased presence of mast cells in periportal regions upon bile duct ligation. The inhibition of mast cell activation with cromolyn sodium inhibits the biliary hyperplasia induced by the bile duct ligation [35]. In another study, the heightened presence of mast cells around bile ducts is associated with an enhanced ductular reaction in diet-induced fatty liver. Depleting mast cells results in reduced liver damage and steatosis in these MASLD mice [28]. In other studies, reduced hepatic fibrosis and biliary hyperplasia within the liver are observed through the use of mast cell-deficient mice or by inhibiting histamine release from hepatic mast cells [9,36]. These insights into the paracrine role of mast cells and ductal reaction during liver damage highlight a crucial pathological function triggered by mast cell presence and their communication with cholangiocytes, indicating the potential utility of mast cell stabilizers in treating MASH-related fibrosis and ductular reaction. In summary, our study reveals that the inhibition of hepatic mast cell activation by CBG mitigates MASH-related fibrosis and biliary proliferation, establishing CBG as a mast cell stabilizer.

TGF-β pathways exert a significant influence on HSC activation as well as the progression of hepatic fibrosis and cirrhosis. Interventions targeting the TGF-β/Smad signaling pathways hold potential for antifibrotic therapy [31,37]. CBG reduces the co-localization of TGF-β1 and mast cells, as well as co-localized HSCs and TGF-β1, providing valuable evidence about CBG for reducing fibrosis and ductular proliferation in MASH. Therefore, we propose a model, in which TGF-β1, secreted by mast cells, plays a role in activating the HSCs. This activation subsequently leads to enhanced collagen deposition and ultimately results in fibrosis. Meanwhile, CBG exhibits anti-fibrotic and anti-ductular proliferative effects by inhibiting the activation and proliferation of mast cells via the inhibition of TGF-β1 secretion from mast cells. Nevertheless, in order to fully elucidate the mechanism of CBG’s action, direct evidence is needed to understand which genes directly interact with CBG, therefore affecting the expression of SMAD 3/7 and NF-κB. A schematic picture of potential signaling pathway by which CBG works is illustrated in Figure 5.

## 5. Conclusions

CBG shows similar anti-fibrotic and anti-inflammation properties in both male and female MASH mice. However, it also shows a differential dose response between male and female mice. The high dosage of CBG treatment appears to worsen liver inflammation and fibrosis in both control and MASH male mice [16], while the same high dose of GBG shows anti-inflammation and anti-fibrotic effects in female MASH mice, indicating that male mice are more sensitive to CBG dosage compared to female mice. Furthermore, CBG significantly reduces the liver-to-body weight ratio in male MASH mice but not in female MASH mice, although a reduction trend exists.

Besides this, we find that CBG plays a significant role in modulating T-cells and monocyte-derived macrophages in MASH mice, which was not investigated in our previous research. Notably, we do not observe a significant beneficial effect of CBG in alleviating liver steatosis in MASH female mice, which aligns with previous findings in male mice.

Rodent models, such as those using the MCD diet, do not fully replicate human MASH pathology, particularly the metabolic syndrome components like obesity, insulin resistance, and dyslipidemia. Translating findings from animal models to humans remains a challenge. MASH develops over the years in humans, but our experimental model attempts to mimic this process within weeks. This accelerated timeline may not accurately reflect the gradual progression of fibrosis, inflammation, and metabolic dysfunction seen in patients. The disease varies significantly among patients due to genetic, lifestyle, and environmental factors. Experimental models may not capture this heterogeneity, making it difficult to generalize findings to the broader patient population. Many MASH patients also suffer from type 2 diabetes, cardiovascular disease, or other metabolic disorders.

In summary, this study investigates the effects of CBG at two distinct doses, with the low dose being ten times more diluted than the high dose. An additional testing of intermediate dosages is needed to gain a clearer understanding of the optimal CBG dosage regimen for potential use in clinical studies. Additionally, the gender-specific response to the high CBG dose will serve as a valuable foundation for future dose-dependent studies aimed at better understanding the therapeutic potential of CBG in addressing MASH-related liver damage. The results of our study represent a significant step toward clinical trials and provide potential avenues for exploring drugs with similar properties to CBG in reducing MASH-related toxicity, potentially lowering the incidence of liver diseases and cirrhosis associated with MASH.

## Figures and Tables

**Figure 1 nutrients-17-01524-f001:**
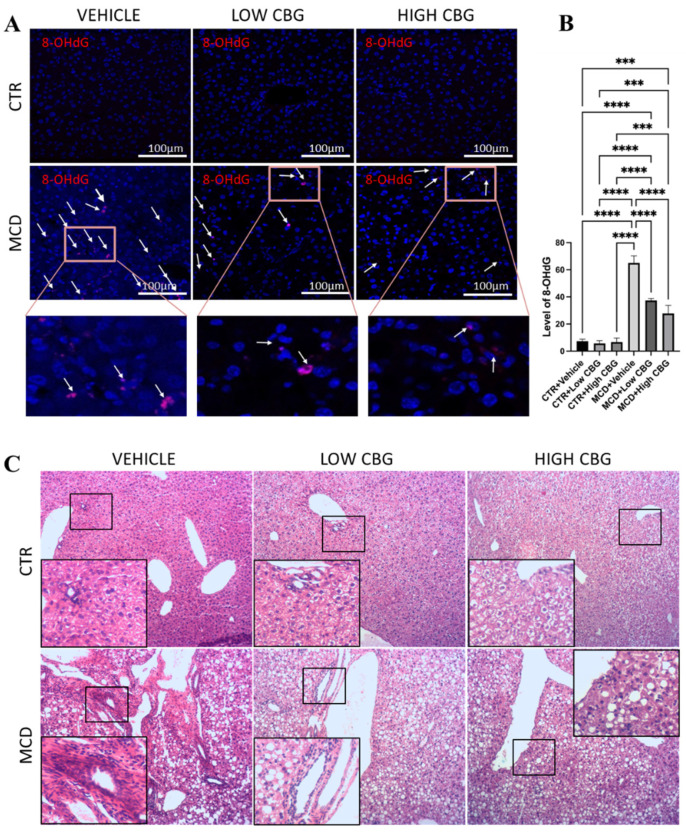
An assessment of oxidative stress and leukocyte infiltration in liver tissue. (**A**) Representative images of the immunofluorescence staining of 8-OHdG of the liver sections from mice of each group. White arrows indicate the overlap of DAPI (Blue) and 8-OHdG (Red). Original magnification ×200. (**B**) The quantification of the level of oxidative stress. *** *p* < 0.001; **** *p* < 0.0001. (**C**) Representative images of H&E staining to observe leukocytes in FFPE liver sections from all groups of mice, *n* = 3 in each group. Hematoxylin stains the nucleus of the leukocytes as blue in color, indicating inflamed areas in liver tissue; eosin stains the remaining part of the liver tissue as pink. Original magnification ×100.

**Figure 2 nutrients-17-01524-f002:**
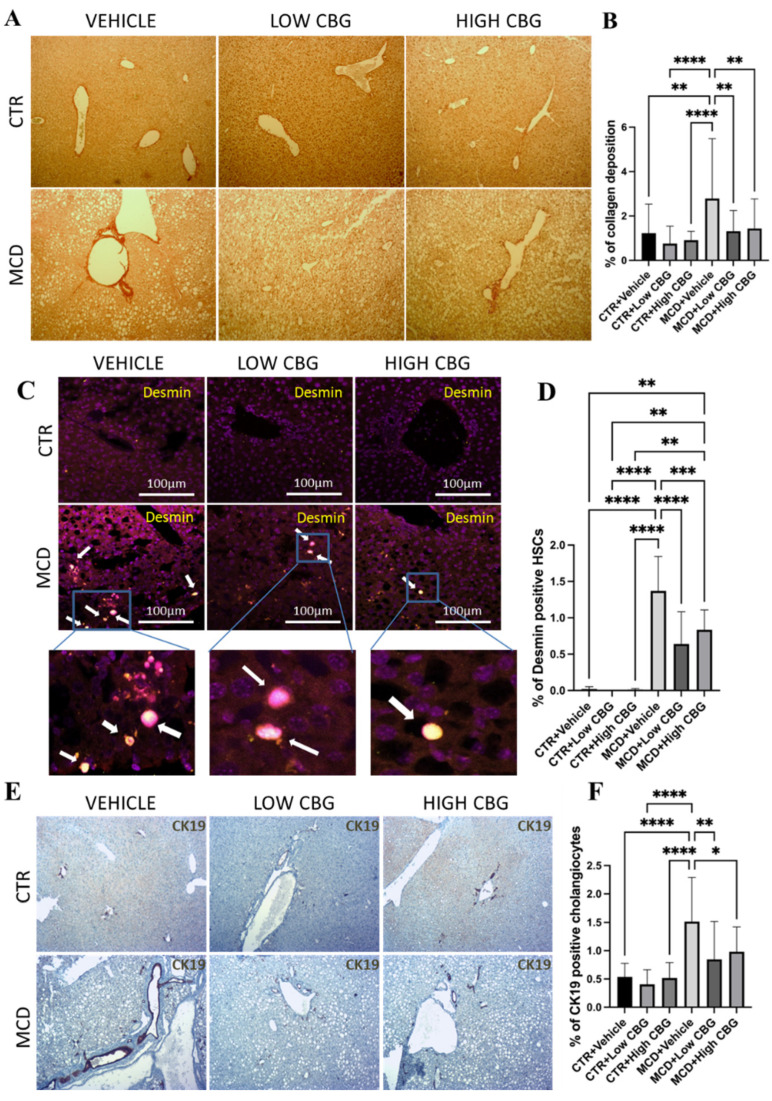
An evaluation of liver fibrosis and ductular proliferation. (**A**) Picrosirius red staining shows collagen deposited in FFPE liver sections. Red intensity represents the deposited collagens indicative of fibrotic liver tissue scars. Original magnification ×100. (**B**) The quantification of the level of fibrosis. (**C**) Illustrative pictures depicting desmin immunofluorescence staining on liver sections from mice in each experimental group. White arrows indicate the overlap of DAPI (violet) and desmin (yellow). Original magnification ×200. (**D**) The quantification of the level of desmin-positive HSCs. (**E**) Representative images of the immunohistochemistry staining of CK-19 (brown) positive cholangiocytes of the liver sections from mice of each group. Original magnification ×100. (**F**) The quantification of CK-19-positive cholangiocytes. * *p* < 0.05; ** *p* < 0.01; *** *p* < 0.001; **** *p* < 0.0001. Quantification is completed using ImageJ software.

**Figure 3 nutrients-17-01524-f003:**
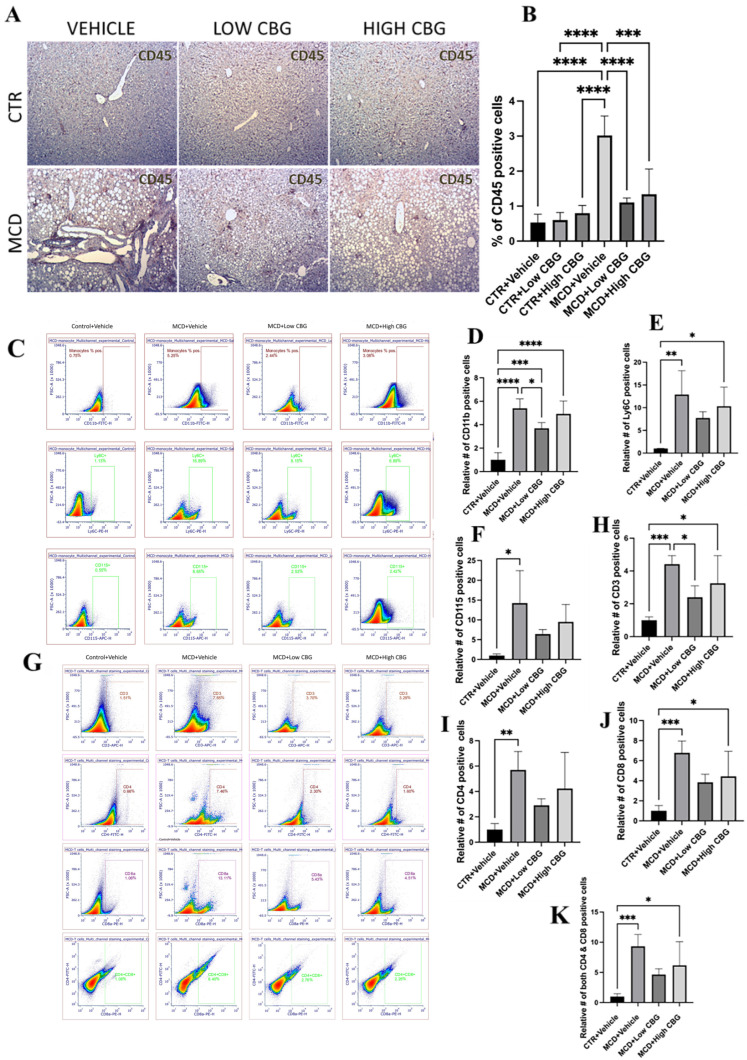
Hepatic inflammation in MASH mouse model with CBG treatment. (**A**) Representative images of immunohistochemistry staining of CD45 (brown)-positive white blood cells of the liver sections from mice of each group. Original magnification, ×100. (**B**) The quantification of the levels of inflammation. Quantification is performed using ImageJ software. (**C**–**F**) Representative images (**C**) and graphical representation of the relative number of CD11b^+^ (**D**), Ly6C^+^ (**E**), and CD115^+^ (**F**) cells in isolated NPCs of the liver from each group of mice. (**G**–**K**) Representative images (**G**) and graphical representation of the relative number of CD3^+^ (**H**), CD4^+^ (**I**), CD8^+^ (**J**), and cells CD4^+^/CD8^+^ (**K**) in isolated NPCs of the liver from each group of mice. All flow data are normalized relative to the average values of the control groups. * *p* < 0.05; ** *p* < 0.01; *** *p* < 0.001; **** *p* < 0.0001.

**Figure 4 nutrients-17-01524-f004:**
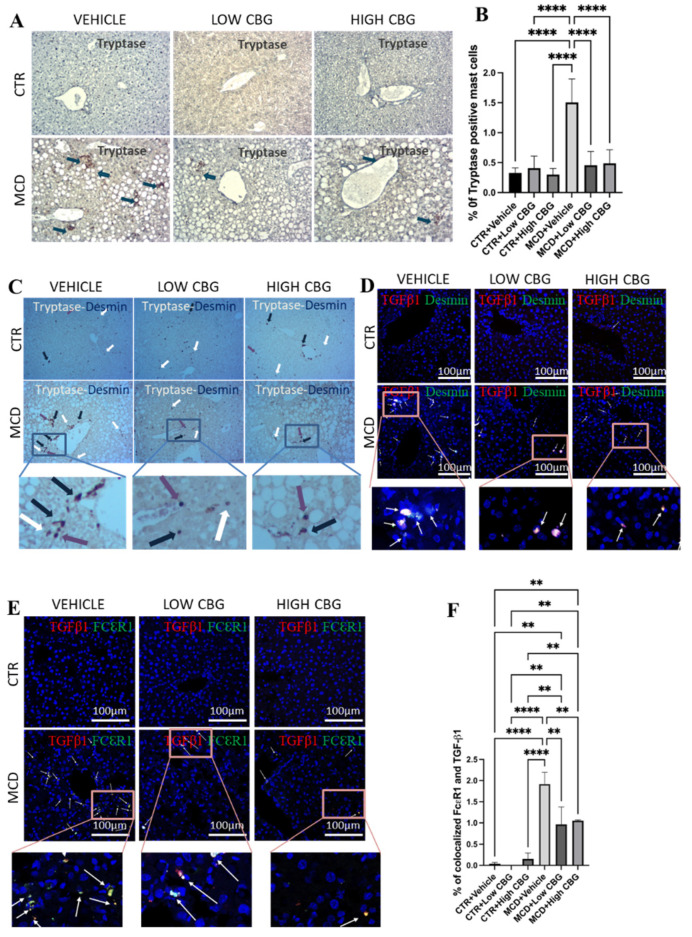
The infiltration of mast cells and crosstalk with HSCs via TGF-β1. (**A**) Representative images of the immunohistochemistry staining of beta tryptase (Tspb1) of the FFPE liver sections from mice of each group. Tspb1-positive mast cells (Brown) are indicated by blue arrows found near the portal triad. The cell nuclei are counterstained with hematoxylin. Original magnification, ×200. (**B**) The quantification of the level of tryptase-positive mast cells. (**C**) Representative images of the co-staining of tryptase and desmin with the immunohistochemistry staining of the FFPE liver sections from mice of each group. Tryptase-positive mast cells (brown) are indicated by white arrows, desmin-positive HSCs (purple) are indicated by purple arrows, and colocalized mast cells and HSCs are indicated by blue arrows. All three are found near the portal triads. Original magnification, ×200. (**D**) Representative images of the immunofluorescence staining of co-localized desmin and TGF-β1 in the frozen liver sections from mice of each group. White arrows indicate the overlap of DAPI (blue), desmin (green), and TGF-β1 (red), highlighting that TGF-β1 secreted from mast cells further activates hepatic stellate cells near the portal triads to promote fibrogenesis. (**E**) Representative images of immunofluorescence staining for the colocalization of FcεRI and TGF-β1 in the frozen liver sections from mice of each group. DAPI stains the nucleus of the cells. White arrows indicate the overlap of DAPI (blue), FcεRI (green) and TGF-β1 (red), meaning that TGF-β1 is secreted from the mast cells. Original magnification, ×200. (**F**) The quantification of the level of co-localized FcεRI and TGF-β1. ** *p* < 0.01; **** *p* < 0.0001. Quantification is performed via ImageJ software.

**Figure 5 nutrients-17-01524-f005:**
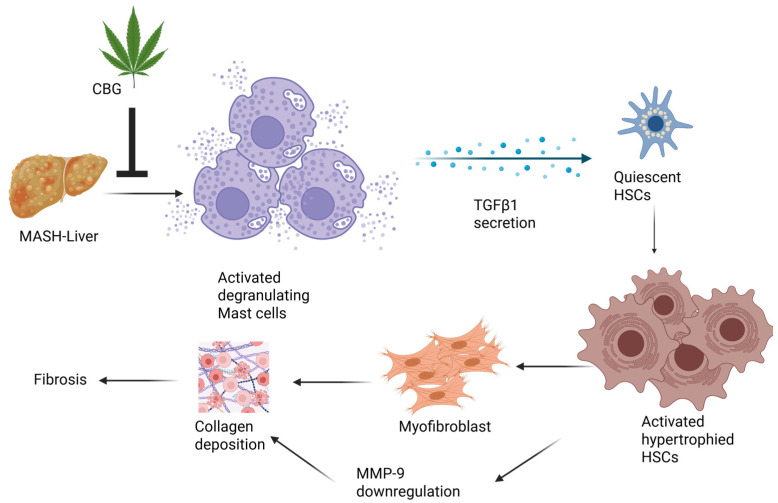
Schematic illustration of potential signaling pathways by which cannabigerol works.

## Data Availability

All original data are available upon request. The data is not shared publicly due to being a part of an ongoing study.

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
