# Peer review of "Cannabigerol Alleviates Liver Damage in Metabolic Dysfunction-Associated Steatohepatitis Female Mice via Inhibition of Transforming Growth Factor Beta 1"

_nutrients, 2025, doi:10.3390/nu17091524_

Round 1
Reviewer 1 Report
Comments and Suggestions for Authors
The text below is a review for a manuscript entitled “Cannabigerol Alleviates Liver Damage in Metabolic Dysfunction-Associated Steatohepatitis Female Mice via Inhibition of Transforming Growth Factor Beta 1”.
The manuscript aims to test the impact of cannabigerol treatment in alleviating the risks related to methionine-choline deficient diet induced metabolic dysfunction-associated steatohepatitis in female mice.
I have several minor comments:
Page 2, at the end pf the first paragraph: “Therefore, there is an urgent need to explore drugs or supplements that can alleviate liver damage in MASH patients.” is missing reference. It is the same with paragraph 2 on page 2.
The names of the molecules, such as Cannabidiol are written with small caps, unless they stand at the beginning of the sentence.
When mentioned for first time Cannabis sativa you should include the name of the authority, therefore it should be Cannabis sativa L. Further you can use the abbreviation C. sativa.
To my opinion table 1 is more appropriate to be as a supplementary.
In material and methods section the text font and size seems to be different and must be unified.
To my opinion, a suitable conclusion is missing.
Comments on the Quality of English Language
The quality of English language is sufficiently enough.
Author Response
The text below is a review for a manuscript entitled “Cannabigerol Alleviates Liver Damage in Metabolic Dysfunction-Associated Steatohepatitis Female Mice via Inhibition of Transforming Growth Factor Beta 1”.
The manuscript aims to test the impact of cannabigerol treatment in alleviating the risks related to methionine-choline deficient diet induced metabolic dysfunction-associated steatohepatitis in female mice.
I have several minor comments:
Page 2, at the end pf the first paragraph: “Therefore, there is an urgent need to explore drugs or supplements that can alleviate liver damage in MASH patients.” is missing reference. It is the same with paragraph 2 on page 2.
Thank you for the comment, we have added the reference to the text. (Line 51, reference 5)
The names of the molecules, such as Cannabidiol are written with small caps, unless they stand at the beginning of the sentence.
Thank you for the comment, we have checked the manuscript and change into small caps.
When mentioned for first time Cannabis sativa you should include the name of the authority, therefore it should be Cannabis sativa L. Further you can use the abbreviation C. sativa.
Thank you for the comment, we have changed it accordingly. (Line 70)
To my opinion table 1 is more appropriate to be as a supplementary.
Thank you for the comment, we have changed table 1 to supplemental table 1
In material and methods section the text font and size seems to be different and must be unified.
We have unified the text font and size.
To my opinion, a suitable conclusion is missing.
Thank you for the suggestion, a new session conclusion is now included in the text. (from Line 512)
Reviewer 2 Report
Comments and Suggestions for Authors
Thank you for the opportunity to review this manuscript. The authors did many assays, and the manuscript is interesting. However, I have some comments. All my questions are in the attachment.

Reviewer 3 Report
Comments and Suggestions for Authors
This manuscript can’t be processed further until the authors solve the main issue of similarities with other published works. 71% similarity index is unacceptable! Take into consideration that this manuscript has 67% of similarities with only one publication: https://digscholarship.unco.edu/cgi/viewcontent.cgi?params=%2Fcontext%2Ftheses%2Farticle%2F1354%2F&path_info=Raznin_Joly_Thsesis_Fall__2023.pdf
Based on these premises, it is impossible to fairly assess this manuscript.
The abstract exceeds the word limit. Some directions for future investigations should be pointed out at the end of the Conclusions.
The references are not formatted according to the journal’s guidelines.
Line numbering, sections and subsections numbering are missing.
Regarding ethics, the authors have to indicate the approval date.
Why did you conduct this study in mice? How can it be relevant for humans? Clarify this in the introductory section.
The histological staining needs to be better explained. What do you mean with this: “These pictures were analyzed using Image J software.”? Why is this in italics?
The same, why is this in italics? (Table 1 for antibody list).
The size and quality of the Figures need to be increased and improved.
In the Discussion, please elaborate on the possible extrapolation of the results to humans.
The study’s strengths and limitations must be addressed in the Discussion.
A Conclusions section and Future Perspectives are missing.
Author Response
This manuscript can’t be processed further until the authors solve the main issue of similarities with other published works. 71% similarity index is unacceptable! Take into consideration that this manuscript has 67% of similarities with only one publication: https://digscholarship.unco.edu/cgi/viewcontent.cgi?params=%2Fcontext%2Ftheses%2Farticle%2F1354%2F&path_info=Raznin_Joly_Thsesis_Fall__2023.pdf
Thank you for your response. The 71% similarity is due to an unpublished thesis from the same research group. This manuscript is original and has not been published anywhere else. Nevertheless, we have modified the text, and now the similarity has reduced to 36% from the cited thesis.
Based on these premises, it is impossible to fairly assess this manuscript.
The abstract exceeds the word limit. Some directions for future investigations should be pointed out at the end of the Conclusions.
Thank you for your response, we have modified the abstract to make it within the word limit.
The references are not formatted according to the journal’s guidelines.
We have changed that to journal’s guidelines.
Line numbering, sections and subsections numbering are missing.
Thank you for the feedback. We have added the line numbering and sections, subsections numbering.
Regarding ethics, the authors have to indicate the approval date.
The IACUC protocol and approved date has sent to the editor.
Why did you conduct this study in mice? How can it be relevant for humans? Clarify this in the introductory section.
Thank you for the suggestion. We have added into the introduction (page 2, line 52-56)
The histological staining needs to be better explained. What do you mean with this: “These pictures were analyzed using Image J software.”? Why is this in italics?
The histological staining was semi-quantified by a software called Image J, which could evaluate the positive staining area. The format issue has been fixed, now it is not italics.
The same, why is this in italics? (Table 1 for antibody list).
We have moved this table as supplemental table 1 as suggested by reviewer 1. And have changed the format based on the journal requirement.
The size and quality of the Figures need to be increased and improved.
Original high resolution figures have been provided in a separate documents.
In the Discussion, please elaborate on the possible extrapolation of the results to humans.
Thank you for the suggestion. This has been added into the conclusion from line 536 to 542.
The study’s strengths and limitations must be addressed in the Discussion.
Thank you for the suggestion. The strengths (line 536-545) and limitations (line 525-535) have been addressed in the conclusion.
A Conclusions section and Future Perspectives are missing.
Thank you for the suggestion. The conclusion session has been added from line 512 to line 546. Future perspective has been added into the document in the discussion (line 444-445, line 455-457, line 475-477, line 506-508, ).
Round 2
Reviewer 3 Report
Comments and Suggestions for Authors
The similarities of this manuscript with other publications are still unacceptable. Anyway, the decision to further process this work will belong to the academic editor and the journal. I don’t agree with its publication in its current form.
Regarding ethics, the authors have to indicate the approval date in the manuscript and not only send it to the editor.
Author Response
Dear Reviewer,
I have added the ethics statement in line 571 to line 574.
Thanks